# Sofigait—A Wireless Inertial Sensor-Based Gait Sonification System

**DOI:** 10.3390/s22228782

**Published:** 2022-11-14

**Authors:** Dagmar Linnhoff, Roy Ploigt, Klaus Mattes

**Affiliations:** 1Institute of Human Movement Science, University of Hamburg, 20148 Hamburg, Germany; 2BeSB GmbH Berlin, 12203 Berlin, Germany

**Keywords:** acoustic feedback, sonification, gait patients, knee angle, sound design, measurement system comparison

## Abstract

In this study, a prototype of an inertial sensor-based gait sonification system was tested for the purpose of providing real-time gait feedback on the knee angle. The study consisted of two parts: (1) a comparison of the knee angle measurement to a marker-based 3D optical capturing system (Vicon, Oxford, UK) with *N* = 24 participants and (2) an evaluation four different sonification feedback versions in an accentuation × pitch (2 × 2) design on a sample of *N* = 28 participants. For the measurement system comparison, the RMSE was 7.6° ± 2.6° for the left and 6.9° ± 3.1° for the right side. Measurement agreement with bias up to −7.5° ± 6.2° (for maximum knee flexion) was indicated by the Bland–Altmann Method. The SPM revealed significant differences between both measurement systems for the area 45–90% (*p* < 0.001) (left) and the area between 45% and 80% (*p* = 0.007) (right). For the sonification perception, the variation of pitch had a significant effect on the perception of pleasantness of the sound. No effect was found for the accentuation of the swing or stance phase.

## 1. Introduction

The use of technology-based real-time feedback has received growing interest in gait therapy to date. Different technical devices have been developed that transform directly measured biomechanical gait parameters (e.g., acceleration, angular velocity, ground reaction forces, and step length) into either visual, acoustic, or tactile real-time feedback [1]. The use of acoustic feedback is especially interesting for gait therapy [2] because it does not interfere with visual information processing, is rich in information, and is intuitive [3,4]. Transforming biomechanical gait parameters into acoustic information has been applied to prevent running injuries or to improve running economy, to improve gait in Parkinson’s disease, and to relearn gait after joint replacement [5,6,7,8,9,10,11,12,13,14,15,16].

In terms of acoustic feedback, there are several types to distinguish in regard to the feedback type and given information. A common form is the use of warning signals or alarms also referred to as event-based sonification [17,18]. For example, in partial weight-bearing training, a beep tone appears whenever a predefined loading is exceeded [19] or in running if tibial acceleration exceeds a predefined boundary value [20]. Moreover, acoustic signals can also be used for reinforcement, e.g., a beep tone whenever a predefined goal is reached [11,21]. This way, the use of acoustic signals implies valence information as “correct” or “wrong”. Another recent acoustic feedback approach is to translate the movement itself into a sound by using movement sonification [22,23,24]. The idea is to transform data relations into a sound without information loss [25,26]. This sonification approach is called parameter mapping and differs from event-based sonification (such as alarms or warning signals) in its continuous and valence-free nature [18,27]. This is what we refer to as sonification here.

Gait is of a rhythmic nature [28], and auditory perception is highly sensitive to rhythm and time [24,29]. From a movement execution perspective, mirroring the spatiotemporal movement structure by sonifying gait data is therefore interesting. This way, sonification can function as an additional (acoustic) reafference of movement execution [30,31]. Applied to gait therapy, the sonification of gait data could help us to understand and correct the gait pattern intuitively and may therefore enhance the learning process. In order to use gait sonification as feedback in gait therapy, some aspects need to be considered, such as which gait data should be used, what the sound should be like, and how it can be accessible to patients and therapists.

In sonification, information can be transported by varying the pitch, volume, or timbre of a tone [32]. On one hand, sufficient information about the actual gait performance would be necessary to correct the gait pattern, but on the other hand, an information overload should be avoided. The trade-off between richness in information and information overload depends on the experience of the user with the acoustic feedback [33]. This raises the question of which gait data would provide sufficient information. Some plausible options are ground reaction forces, the velocity of segments (e.g., tibial acceleration), or joint angle trajectories within a gait cycle (e.g., hip, knee, or ankle angles). If the feedback concerns the symmetry of the gait pattern, feedback would be required for both legs, increasing the amount of information. Overall, the acoustic information should depend on the goal of the gait intervention and the processing capacity of the receiver.

Another important consideration related to the user or listener experience is aesthetics and pleasantness of sound [34]. The emotional or affective state influences the motivational state [35], the ability to process information, and the perception of a given task as hard or easy [36,37]. The perception of a tone as pleasant or unpleasant is, among other things, related to the volume, pitch, and duration. Studies have shown that complex interactions exist between the perception of pitch (as high or low) and the loudness of auditory feedback, and that there is an opposing relationship between emotional and informational perception [32,38]. Therefore, when considering which information should be conveyed via which acoustic dimensions, the influence of the sound on the affective state of the learner should also be taken into account.

For the technical implementation of sonification feedback in gait therapy, accurate real-time measurement and translation of the corresponding gait parameter into a sound must be realized. Nevertheless, the technique should be cost-effective and simple in application in order to make it accessible to patients and therapists. Inertial sensor-based systems are easy to transport, and compared to optoelectronic camera-based systems (e.g., Vicon, Oxford, UK), they are low in cost, while still having good measurement agreement depending on the gait parameter [39,40]. These systems usually integrate multiple inertial measurement units (IMUs or MIMUs if a magnetometer is included) that provide position and acceleration data. Based on this data, various gait parameters (e.g., step length, segment angle, and segment velocity) are computed as output. This output can also be directly converted into acoustic real-time feedback, which has been previously realized in other settings [7,8,10,41,42,43].

The aim of this study was to test and evaluate a newly developed MIMU-based gait sonification system (sofigait, BeSB, Berlin, Germany). Sofigait translates sagittal knee angle kinematics (as the gait data) into a tone whose pitch and volume vary according to the course of the curve (parameter mapping sonification). The sagittal knee-angle curve reflects the division into a stance phase and a swing phase, as both are initiated by a minimum angle value, followed by an increase of the value to about 18° (stance) and 65° (swing) in the middle of the phase, followed by a decrease, at whose lowest point the other phase begins [28]. Thus, it should be possible to perceive spatiotemporal asymmetries (e.g., limp) by listening to both legs simultaneously.

The study evaluated the concurrent criterion validity of the measurement and the perception of the feedback sound during asymmetric walking. Consequently, it consisted of two parts: (1) a comparison of our inertial sensor-system measured knee-angle kinematics to a gold-standard optical system (Vicon, Oxford, UK) and (2) testing of four different sonification modes in participants while performing asymmetric walking.

## 2. Materials and Methods

### 2.1. Sensor System (Sofigait)

The sofigait system (BeSB, Berlin) is a prototype of an inertial sensor-based knee-angle measurement system (KMS1) with an integrated sonification module. It consists of four inertial measurement units (technically, they are MIMUs, but are hereafter referred to as IMUs) and a tablet computer (acer Aspire Switch, 2014) for integration and acoustic processing of the data (Figure 1b). The four IMUs (Bosch BNO080) each integrate a triaxial accelerometer, triaxial gyroscope, a magnetometer, and a 32-bit ARM^®^ Cortex™-M0+ microcontroller. Two IMUs are interconnected with a cable and function as one angle measurement unit for one leg (attached to the upper and to the lower leg to measure the knee angle) (Figure 1c,d). The knee-angle data are then fed to an algorithm for sound synthesis (sonification), which realizes the acoustic feedback. The acoustic signal is transmitted via a Bluetooth transmitter (Sennheiser BT T100) to wireless headphones (Sennheiser hd-450-bt) with an overall time delay of fewer than 50 milliseconds.

#### 2.1.1. Components and System Setup

The KMS1 knee-angle measurement system is based on the development of an autonomous battery-driven IMU with a Bluetooth wireless connection (LS1) (Figure 1a). The LS1 integrates a 32-bit ARM microcontroller (MCU) (100 MHz), an IMU, and a Bluetooth module for data computing and transmission. For the power supply, a battery and a charging module are also installed. To measure angles between two segments, the LS1 was extended by an additional IMU to obtain and combine position data from two segments (upper and lower leg). The data of both IMUs are transmitted separately to the computer via Bluetooth.

#### 2.1.2. Data Acquisition and Transformation

Each IMU provides position data in the form of quaternions, as well as gyro and acceleration values with a data rate of 100 Hz, a resolution of 0.1°, and an accuracy of 0.5°, respectively. For the knee-angle measurement, the manufacturer-delivered position data in the form of quaternions are used. The Datasheet of the Bosch BNO080 can be accessed online: https://www.ceva-dsp.com/wp-content/uploads/2019/10/BNO080_085-Datasheet.pdf (accessed on 10 September 2022). The quaternions (sensor orientation data) are obtained via sensor fusion (referred to as “game rotation vector” by the manufacturer) [44]. A temporal drift of 1° per h about the z-axis (perpendicular) was previously measured in the laboratory. The IMUs process motion relative to a frame of reference, which is represented as X-axis horizontal, positive to the right, Y-axis vertical, positive along the face of the device and Z-axis positive toward the outside of the front face of the device (Figure 2a).

First, the sensors are attached to the leg so that they are positioned laterally on the thigh and lower leg and aligned equally along the vertical axis on both legs (Figure 2b). Thus, both IMUs of one pair (or leg) can be assumed to be in the same plane and have the same orientation. The angle calculation is performed on the computer with the self-developed sofigait software (BeSB, Berlin), using the IMU transmitted position data (P_1_ and P_2_) in form of quaternions ([Qw, Qx, Qy, Qz]) [44].
P_1_ = [Qw_1_, Qx_1_, Qy_1_, Qz_1_]
P_2_ = [Qw_2_, Qx_2_, Qy_2_, Qz_2_]

The sensors are initialized by switching them on. At this point, the orientation of each single IMU is still estimated and transmitted to the computer separately. In the next step, the two IMUs of one measurement unit are transformed into a common coordinate system or measuring plane (M). For this purpose, a rotation matrix in form of quaternions (Mrq) is obtained in an iteration process based on the source quaternions of P_1_ and P_2_. The rotation matrix is then multiplied with the source quaternions, resulting in P_1_ = P_2_ and aligned vectors (V1 and V2) for the two IMUs. The vectors are the basis of the angle calculation. The virtual aligned position is set to an angle value (α) of 0° (Figure 2c).

In a further optional initialization step, the sensors are brought into a 90° (perpendicular) position (Figure 2d), and a correction factor is estimated to compensate for rotations along the vertical axis that occur anatomically due to muscle and tissue. This step increases the accuracy of the measurement but can also be skipped if the bending of the knee is not possible.

After the initialization procedure, real-time measurement is processed while the person is walking in any direction of the laboratory as far as the Bluetooth signal still reaches the receiver (up to 8m was possible in trial measurements). The program on the computer also provides a continuous real-time display of the actual measured knee angle trace (displayed as a single line per leg), the rate of data transmission, and the transformed quaternions. The initialization and calibration status of the KMS1 can therefore be monitored permanently. The user is guided through the initializing procedure by a status display in the lower part of the screen. The acceleration (m/s^2^) and gyroscope data (rad/s) of all planes and the knee angle left and right (degree) can additionally be recorded and stored in txt-files.

#### 2.1.3. Sonification

The knee-angle data are sonified by parameter mapping [45]. The sagittal knee angle is measured and sonified according to the usual declaration in the literature, where the extended leg is referred to as 0° and the angle increases with increased flexion (180°-between-segment angle of the thigh and lower leg) (Figure 3).

Four sonification versions are implemented, differing in which gait phase (stance or swing) is emphasized in a high or low version:(1)Soni 1.1: Sine continuous tone, swing phase from 35°.

Logarithmic mapping of the knee angle course from −45° to +90° to frequencies from 220 Hz to 1760 Hz, so that at 0° the concert pitch with 440 Hz results and thus modulates a sine oscillator.

In addition, the knee angle was mapped to the volume so that the continuous tone below 35° fades out linearly to 0°, resulting in an alternating left/right sound (for schematic illustration see Figure 2).

(2)Soni 1.2: Sine continuous tone, swing phase from 35°

Same principle as Soni 1.1, but a half octave lower.

(3)Soni 2.1: Sine continuous tone, stance phase up to 35°.

The same principle as for 1 and 2, but in such a way that the volume of the continuous tone decreases linearly from 0° to 35°. This results again in an alternating left/right sound, but with emphasis on the smaller angles.

(4)Soni 2.2: Sine continuous tone, stance phase up to 35°.

Same principle as Soni 2.1, but a half octave lower.

### 2.2. Study Protocol

#### 2.2.1. Participants

The study was conducted with a sample of *N* = 28 healthy participants (15 females and 13 males, aged 22.8 ± 2.9 years). Four of the twenty-eight participants took only part in the sound evaluation part of the study and missed out on the measurement comparison.

#### 2.2.2. System Check

Before the study, the sofigait system was tested on two female participants (ages 32 and 36, and heights 172 cm and 170 cm, respectively). Each did five trial walks with a speed of 4 km/h for five minutes while sofigait constantly measured both knee angles. Within these five minutes, the gait data were captured and stored for 10 s every minute to check for drift by time. After each walk, the system was recalibrated to check whether there would be offsets between the repeated measures (results: Appendix B).

#### 2.2.3. Participant Rating

We developed a short rating to evaluate the feedback perception (Appendix A). The questionnaire consisted of eight ratings on a 6-point Likert scale, from the worst to the best rating. The even number was chosen to avoid rating the middle in case of unsureness. The conception and the rating procedure were based on basic principles of the evaluation of auditory displays [33]. For example, the ratings were phrased short and unidimensional because of perceptual limitations for auditory stimuli. Of the eight ratings, three ratings (on gait reflection, perception of asymmetry, and left-right separation) were averaged for the information dimension, and one rating (whether the sound was perceived as “unpleasant” or “pleasant”) formed the other dimension (pleasantness). The remaining four ratings served as additional information but were not analyzed for this study.

#### 2.2.4. Study Procedure

The study protocol consisted of a first part in which the participants walked on the treadmill at 4 km/h speed for 20 s while their gait was simultaneously recorded by the sofigait sensor system (BeSB, Berlin) and Vicon (Vicon Motion Systems Ltd., Oxford, UK). The Vicon data were acquired and processed by using the lower body Plug-in Gait model (PiG) and Nexus software version 2.12.0 (Vicon Motion Systems Ltd., Oxford, UK). The same ten gait cycles captured by each system were analyzed for the measurement system comparison. After completion of this study part, the optical markers were removed, and the participant was fitted with a knee brace (Novamed 6540) that restricted knee flexion on the right leg. The aim was to artificially create a gait asymmetry in each participant for the second study part. Subsequently, each participant walked five consecutive times on the treadmill, with a reduced speed of 3.5 km/h to compensate for the gait restriction. For each walk, acoustic feedback generated by sofigait was provided via wireless headphones (Sennheiser hd-450-bt). Four different feedback versions (described in Section 2.1.3) were randomized in trials 2–5, while the first trial served as an acclimatization trial. After each walk, the rating sheet (Section 2.2.3) was handed out to rate the perception of the feedback. The participants were not introduced to the ratings beforehand, nor were they told what exactly they should pay attention to while hearing the feedback. They were only told that they would have to rate how they perceived the feedback directly after each walk. They also knew that their first walk was a familiarization trial and would be repeated in the following four walks. In this way, the first trial served to familiarize them with the procedure and let them know what to pay attention to, without the attention being directed by the instructor.

### 2.3. Data Analyses

Out of 24 datasets for the measurement comparison, 22 could be included in the final analysis. The reason for the exclusion of two datasets was that the data collection by the sofigait system had not worked, and therefore the data were not available. The 22 datasets included 11 males (height: 1.83 m ± 0.06 m; weight: 78.5 kg ± 13.9 kg) and 11 females (height: 1.70 m ± 0.5 m; weight: 64 kg ± 9.3 kg). The knee-angle data of both systems recorded with 100 Hz were transferred into Excel sheets for further processing with Python. With a self-written Python routine (Python code: Appendix A), the following steps were executed for statistical analysis:(1)Filtering of the sofigait data (Butterworth 2nd order, low pass, cutoff 6 Hz);(2)Extracting 10 step cycles and cut them into 10 single curves;(3)Computing the mean of all 4 events per system and side;(4)Normalizing all curves to 100 data points;(5)Computing mean curves (1 curve per side per system);(6)Exporting the mean curves and mean events;(7)Reshaping exported curves into four matrices (*N* × data points) for each system and side;(8)Statistical comparison of curves.

The root-mean-square error (RMSE) of the knee-angle curves between sofigait and Vicon was calculated for each participant (Equation (1)), where *N* is the number of data points (100). A mean RMSE was calculated for the 22 individuals for each sensor/side respectively. Furthermore, a discrete analysis was performed for four events in a standardized 100% gait cycle: the maximum of knee extension at the beginning of a gait cycle (Min 1), the maximum of flexion in the stance phase (Max 1), the maximum of extension in inertial swing (Min 2), and the maximum flexion in the swing phase (Max 2) (Figure 4). The mean angle values of the events (Min 1, Max 1, Min 2, and Max 2) per dataset (Step 3) were imported into SPSS (IBM statistics, version 27), and the absolute differences between both measurement systems were then assessed by Bland–Altman Analysis [46]. For the continuous comparison of the curves’ shapes, a two-group *t*-test-based one-dimensional SPM [47] was performed for each sensor/leg-side separately, with an α-level of 0.05. Additionally, the left side and right side for each system were compared via paired *t*-test-based SPM with an α-level of 0.05 (Appendix A).
(1)$$RMSE=\sqrt{\frac{1}{N}\overset{N}{\underset{k=1}{\sum }}{\left(angle\;sofigait-angle\;Vicon\right)}^{2}}$$

The participant ratings (second part) were transferred into SPSS, and mean ratings for the two perceiving dimensions (information and pleasantness) were obtained. The data were first checked for normal distribution by using the Shapiro–Wilk Test. Then the Friedmann Test was conducted for statistical comparison between the four sonification versions and a 2 × 2 ANOVA (accentuation × pitch) to identify the influence of the separate factors on each perceiving dimension and test for interaction between the factors.

## 3. Results

### 3.1. Measurement System Comparison between Vicon and Sofigait

#### 3.1.1. Root-Mean-Square Error

The mean RMSE was 7.6° ± 2.6° for the left and 6.9° ± 3.1° for the right side, respectively, ranging from 3.8° to 13.06° for the left and 3.1° to 14.54° on the right side (Table 1).

#### 3.1.2. Comparison of the Main Events between Vicon and Sofigait

The Bland–Altmann Plots revealed measurement differences being within the upper and lower limits of agreement (±2 SD of the mean difference) for Min 2 and Max 2 and one value being out of boundaries for Min 1 and Max 1 (Figure 5). The bias (average difference between Vicon and sofigait) were 3.5° ± 4.2° for Min 1, 1.2°± 4.6° for Max 1, −1.6 ± 4.7° for Min 2, and −7.52 ± 6.2° for Max 2. Limits of agreement were in a range between 16.5 ° (from lower to upper limit for Min 1) and 24.3° (Max 2).

#### 3.1.3. Continuous Comparison of the Curves (SPM)

The one-dimensional SPM revealed no difference between the left and the right side/leg in each measurement system respectively (Appendix A). For the continuous comparison between both systems (Vicon vs. sofigait), significant differences were found for the left and the right side, respectively. For the left side, the first and last 3% (*p* = 0.046; *p* = 0.049) and the area 45–90% were significantly different (*p* < 0.001), whereas for the right side, only the area between 45% and 80% was significantly different between both systems (*p* = 0.007) (Figure 6).

### 3.2. Sound Perception

The analysis of the additional gait data collected revealed that maximum knee flexion of the right leg was reduced by an average of 11.8° (19.5%) in participants compared with the left leg. The participants rated feedback version Soni 1.2 highest on both dimensions (information and pleasantness). Soni 2.1 was rated lowest in pleasantness, and version 2.2 was lowest in information (Table 2). The Friedmann Test revealed no significant difference between the four versions in the information dimension (*p* = 0.901), but it did in the pleasantness dimension (*p* = 0.012). The ANOVA revealed a significant effect for the factor pitch on the pleasantness dimension (F = 9.22; *p* = 0.005), but not for the factor accentuation. No pitch × accentuation interaction was found on either dimension.

## 4. Discussion

The aim of this study was to introduce and test a recently developed prototype of a real-time acoustic gait feedback system (sofigait, BeSB Berlin) that uses the sagittal knee-angle motion to synthesize sonification feedback on gait. The primary goal was to evaluate the sonification feedback in vivo on a sample of 28 participants while asymmetric walking. Since the sonification feedback relates to the real-time measured knee angle, we first tested whether the knee angle measured by the sofigait system corresponds to that measured by a gold-standard optical capturing system (Vicon, Oxford, UK) to test for concurrent criterion validity. Subsequently, we evaluated the perception of four different versions of the gait-sonification feedback during asymmetric walking.

With sofigait, we aimed to develop a gait feedback system that is easy to set up, low in cost, and thus accessible to therapists. Since optical marker-based systems are costly and require space, time, and training, these are often not available in clinical settings. As an alternative for joint-angle-measurement electronic goniometers can be used, which are low in cost, portable, and have proven good measurement reliability for the knee joint angle while walking [48,49]. Another inexpensive and easily transportable option is inertial sensors, which are usually a combination of goniometer(s), accelerometer(s), and magnetometer(s) and can therefore be used to measure joint kinematics and spatiotemporal gait parameters. Inertial sensor-based gait measuring systems have proven reliable in measuring various gait parameters [39,40]. Some of these systems are commercially available, including analysis software for use in clinical settings (e.g., Reha Gait, and Hasomed) [50]. However, gait parameters are often calculated after the data have been recorded, and the built-in computing algorithms are not always accessible. The sofigait system was developed by using inertial sensors (Bosch BNO080) so that the raw data supplied by the sensors can be directly processed and sonified, allowing the system to function as a real-time feedback device.

The results show that there are differences between the optical motion capturing system and the prototype sofigait system. The RMSEs of 7.6° ± 2.6° (left) and 6.9° ± 3.1° (right) is comparable to Favre et al. [51], who introduced a functional alignment approach by using two IMUs in a similar position and way. They reported mean differences of 8.1° ± 5.4° for the sagittal knee-angle motion during a walk along a 7 m pathway. Picerno et al. [52] also used IMU devices placed on the upper and lower leg, in combination with a magnetometer-aided anatomical alignment calibration procedure to obtain knee-angle kinematics. They reported a relative RMSE of 1.9° for the sagittal plane movement while walking. However, they aligned the curves to compensate for the absolute offset of 2.4 degrees. Cooper et al. [53] used only acceleration and angular velocity measurement of two IMUs without integrating magnetometer data to measure knee-angle kinematics. With their method, they simplified the knee to be a perfect hinge joint, which resulted in a higher measurement agreement. The authors reported RMSEs of 1.0° ± 0.4° for a walking speed of 3 mph (0.8 km/h faster than our participants). The Bland–Altman Method showed sufficient measurement agreement between both systems. However, the limits of agreement were in a range of up to ±12° because of the relatively high standard deviations of the differences between both systems. This matches the previously reported findings regarding the RMSEs. All of the aforementioned studies had smaller sample sizes (two to eight participants), with the main objective of evaluating the precision of their suggested measurement or calibration procedures for gait diagnosis. Although the main objective here was not to evaluate measurement accuracy for the purpose of gait diagnosis, the results appear to be consistent with previous findings. Therefore, we consider this to be a moderate agreement between both systems, as expected in respect to previous research. Sofigait reflects the spatiotemporal course well, but it overestimates the peak values, especially in the case of maximum knee flexion (Max 2). This is supported by the results of the SPM that revealed that the curves were significantly different for around 50% of the gait cycle (left) and 40% (right), which were the areas around the maximum peak of the curve. For the purpose of sonification, which is to transfer the movement structure (data relations) into a sound that provides information about the relative sequence and size of the peaks, sofigait has shown sufficient concurrent criterion validity.

The second part of the study was to evaluate the sonification perception on a sample of healthy but gait-manipulated participants. Sonification feedback transports information via different dimensions (e.g., volume, pitch, and timbre). These dimensions can be manipulated to enhance the perception of information and/or make the sound more pleasant [32,54]. The basic idea was that the receiver hears gait asymmetries between the right and left leg by the acoustic reflection of the sagittal knee-angle movement. We thus assessed the perception of the gait structure as the first dimension of the self-developed participant rating (information dimension). Moreover, in the case of acoustic feedback, the perceived pleasantness plays a role in motivation and also in information processing [32]. Therefore, the dimension of pleasantness (affective perception) was the second dimension of interest. To investigate the influence of changes in single sound dimensions on these two perceptual dimensions, four different versions of the knee-angle sonification were compared. We created two basic feedback versions, accentuating either the swing or the stance phase, and for each of the two, a high- and a low-pitched version were implemented (see Section 2.1.3), resulting in four different versions, varying in the format accentuation × pitch (2 × 2). As expected, the variation of pitch did significantly influence the affective perception dimension (pleasantness). A lower pitch was perceived as more pleasant. Interestingly, the accentuation of the feedback seemed not to influence the participants’ perception of their gait asymmetry (information dimension). Since the gait manipulation mainly reduced the maximal knee flexion in the swing phase, we expected that swing-phase-emphasized feedback would be rated higher on the information dimension. However, this was only supported by the descriptive data. The idea that a higher pitch may influence information and pleasantness inversely was also not supported by the data. Since the swing accentuated and lower pitched version (Soni 1.2) was rated best on both dimensions (information and pleasantness), this would be the version that we would recommend for patients with restrictions in knee movement. These could be, for example, patients after knee-joint replacement or patients after a cruciate ligament injury.

## 5. Limitations and Conclusions

The sofigait system was developed as an inertial sensor-based gait feedback system that generates real-time sonification feedback on knee-angle kinematics. The overall goal is to implement this type of gait feedback device in gait therapy. A major limitation is that the study was conducted with healthy participants and not with individuals actually suffering from gait limitations. Therefore, conclusions drawn from the second part of the study for gait therapy are limited. The advantage, however, is that motion restriction in the knee was standardized here. It also remains unknown whether other gait parameters are equally or even better suited to generate appropriate gait feedback. Finally, a self-built prototype of an inertial sensor-based system is prone to error due to lack of experience and standardization of measurement procedures. One advantage, however, is that most raw measurement data are accessible, but this is not always the case with commercially offered gait measurement systems. Our results are promising and now need to be transferred from the laboratory to practice. There is still a need for development to improve the application of sofigait, but the measurement accuracy is already sufficient. With regard to sound development, it can be stated so far that one way to influence the affective sound perception is to change the pitch. For information perception, this seems to be more complex and requires further investigation. More research is also needed to draw conclusions about whether and how sonification feedback can help people with different types of gait disorders. Overall, gait sonification is a promising area for future interdisciplinary and applied research on feedback strategies for gait therapy.

## Figures and Tables

**Figure 1 sensors-22-08782-f001:**
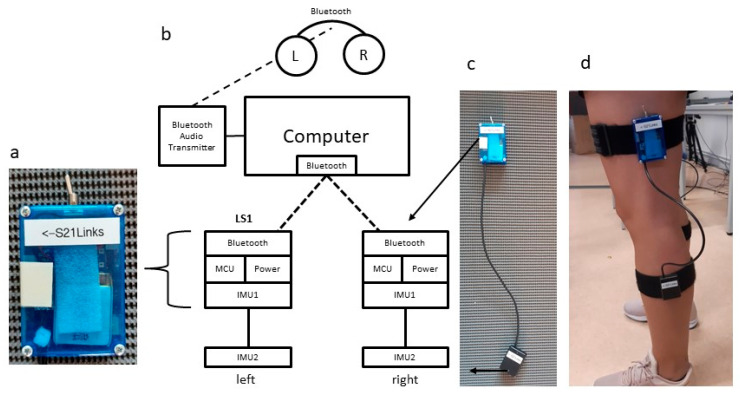
Components of the sofigait system (KMS1). (**a**) LS1 unit consisting of IMU, MCU, Bluetooth module and battery. (**b**) Schematic illustration of the sofigait system. (**c**,**d**) One measurement unit and leg positioning.

**Figure 2 sensors-22-08782-f002:**
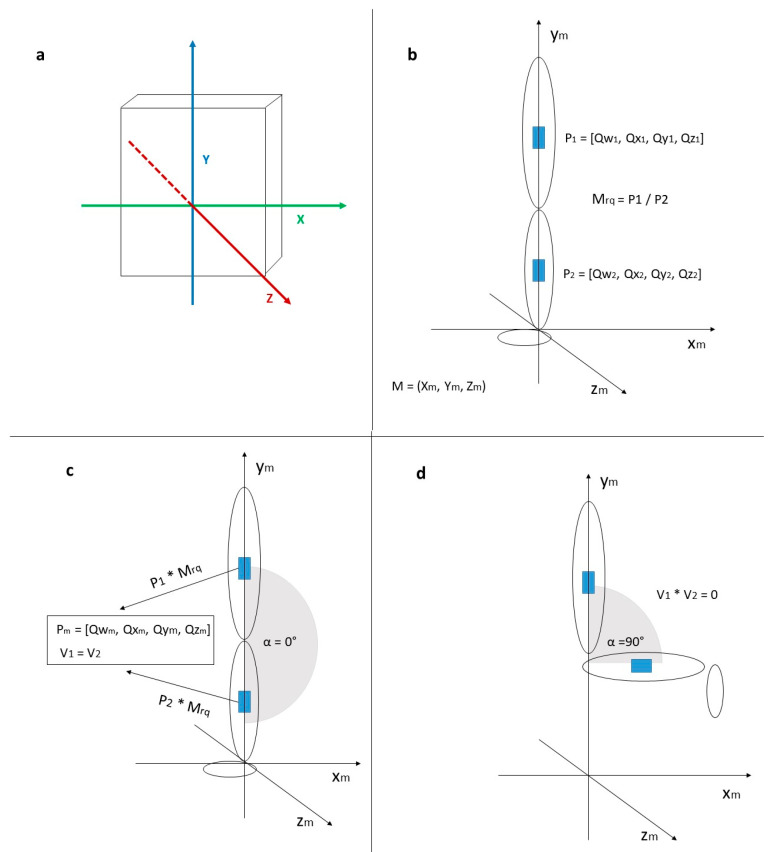
(**a**) Sensor orientation to reference frame according to manufacturer, (**b**) sensors positioned on the leg before initialization procedure. Each sends position data (P_1_, P_2_) separately. (**c**) After initialization, the sensors begin in a common coordinate system after multiplying the position data (P) with the rotation matrix (M_rq_), (**d**) bringing the leg to a 90° position.

**Figure 3 sensors-22-08782-f003:**
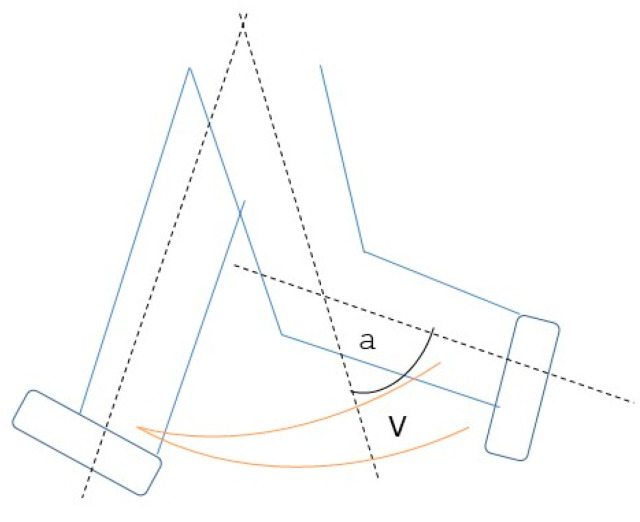
Schematic principle of sonification version Soni 1.1 with V = Volume (increasing with increasement of a = knee angle (knee angle (a) is 180°-between-segment-angle (α)).

**Figure 4 sensors-22-08782-f004:**
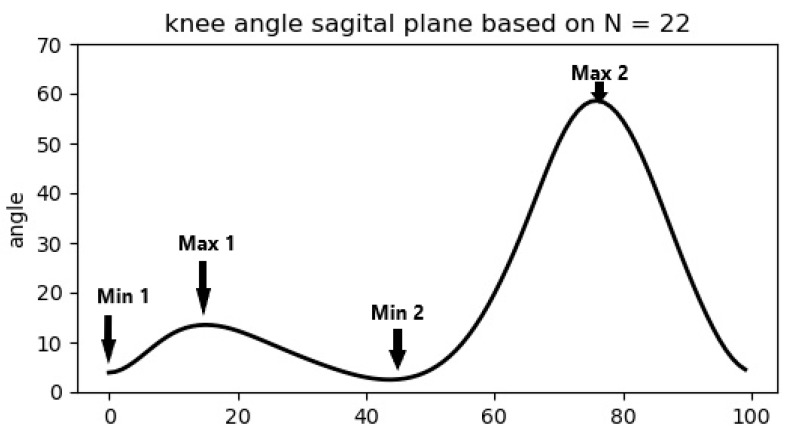
Mean sagittal plane knee-angle curve y-axis = angle in degree (°) in a gait cycle normed to 100% (x-axis) with the four main events as defined and based on the Vicon data of this study (*N* = 22).

**Figure 5 sensors-22-08782-f005:**
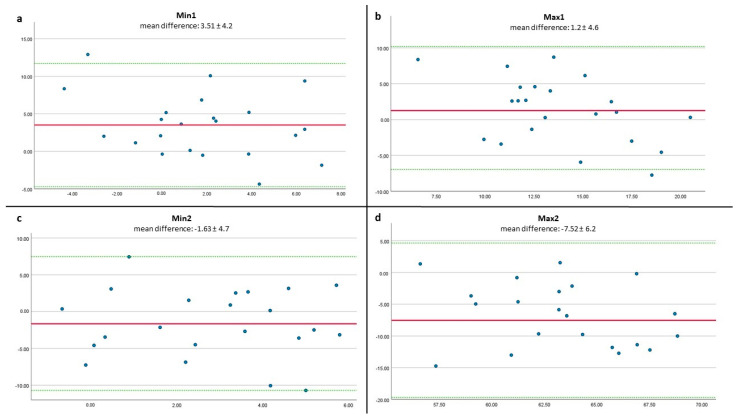
Bland–Altmann Plots x-axis = mean of methods in degree (°), y-axis = difference between both methods in degree (°). The solid line indicates the mean difference (bias) and the dotted lines indicate the limits of agreement for (**a**) Min 1, (**b**) Max 1, (**c**) Min 2, and (**d**) Max 2.

**Figure 6 sensors-22-08782-f006:**
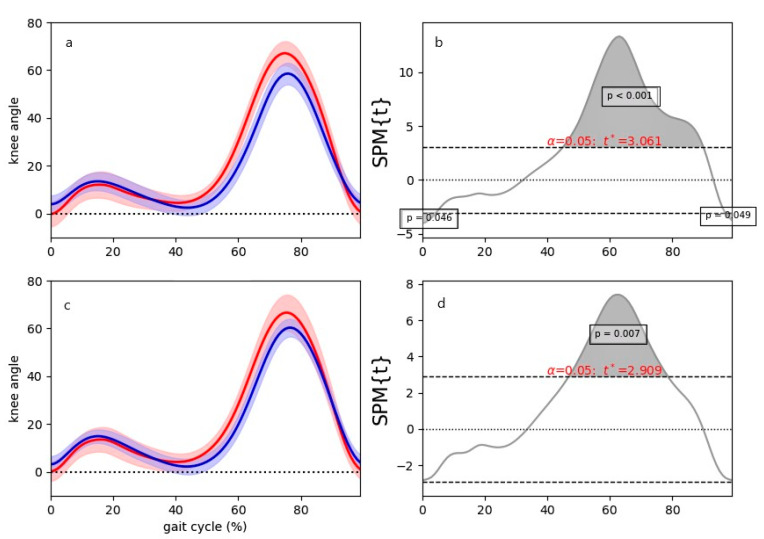
Results of the SPM analysis. (**a**,**c**) Mean curves (solid lines) and SD (shades); x-axis = knee angle, and y-axis = % of gait cycle. (**b**,**d**) SPM results with α level (0.05) based on the t-distribution (dotted lines) and *p*-values; x-axis = t-values, and y-axis = % of gait cycle.

**Table 1 sensors-22-08782-t001:** Root Mean Squared Error for each individual (sofigait vs. Vicon) in degree (*N* = 22 participants).

Participant	RMSE Left	RMSE Right	Participant	RMSE Left	RMSE Right
1	7.35	6.59	12	3.8	6.99
2	6.96	3.10	13	5.85	3.33
3	7.13	5.93	14	4.94	7.18
4	5.95	6.83	15	8.08	7.03
5	13.06	8.88	16	6.11	3.86
6	12.21	14.54	17	8.37	9.66
7	6.33	4.69	18	4.81	5.81
8	7.93	5.29	19	10.53	7.57
9	5.57	6.60	20	5.73	3.47
10	10.11	3.73	21	4.22	11.70
11	4.6	6.06	22	9.78	13.15
			∑	7.6 ± 2.6	6.9 ± 3.1

**Table 2 sensors-22-08782-t002:** Ratings (mean ± SD) for the four feedback versions (Soni 1.1–Soni 2.2) in the two dimensions on a Likert scale of 1–6 (*N* = 28 participants).

Version	Accentuation	Pitch	Dimension I (Information)	Dimension 2 (Sound Perception)
Soni 1.1	Swing	high	4.5 ± 1.1	3.9 ± 1.4
Soni 1.2	Swing	low	4.7 ± 1.2	4.3 ± 1.4
Soni 2.1	Stance	high	4.4 ± 1.2	3.4 ± 1.3
Soni 2.2	Stance	low	4.3 ± 1.3	3.8 ± 1.6

## Data Availability

The datasets presented in this study are available upon request from the corresponding author.

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
