# Peer review of "Sofigait—A Wireless Inertial Sensor-Based Gait Sonification System"

_sensors, 2022, doi:10.3390/s22228782_

Round 1

Reviewer 1 Report

I found the paper well-written, easy to comprehend and of an interesting topic. However, I do not think that you had applied a sufficiently rigorous approach to evaluate your research question 1 (please see specific comments below). I would encourage you to rephrase the goal to RQs 2 and 3 only as those are the ones you provided convincing evidence for.

Methods 

lines 130-133: Unclear whether the MEMS manufacturer-provided quaternions were used, or if you computed the orientation quaternions yourself, please clarify. Also BNO080 datasheet indicates that the device has a magnetometer, which can be used to obtain a geomagnetic rotation quaternion. Which rotation quaternion did you use if you used the manufacturer-provided quaternion?

lines 136 - 142: I cannot work out how your calibration works, please provide sufficient details to enable replicating. Evaluating 3D kinematics using a pair of IMUs requires knowing the laboratory orientation of both of the IMUs at all time points. This is "easily" accomplished if MIMUs are utilised (i.e., a magnetometer is also included) whereby standing in the anatomical orientation is used to figure out the MIMU to body segment orientation, and thereafter using segment inertial signals i.e., rotate sensor recordings with the sensor to segment quaternion. Since this is started from the anatomical pose, the segment coordinate system headins are aligned. Indeed, Opensim uses this approach for the MIMU-based kinematics. Without a magnetometer one would need to establish the sensor to segment quaternion by using a calibration procedure (I think that this is what you have done), and thereafter start calculations from the anatomical pose. However, just a single rotation will not enable establishing the sensor to segment alignment uniquely (the system of equations remains under determined), at least three non collinear rotations are required for a fully determined solution. The comment is conceptually speaking, I did not revisit the exact formulation when writing this comment. 

On a related note, perhaps you could also add information on your QA procedures to ensure 1 deg/h temporal drift. Gyroscope drift varies over time -> session by session calibration  + ample warm-up time (around 10 min) for the electronics is often applied.

lines 193 - 199: It appears that you had N = 2, measured in a single measurement session (5 x 5 min walk each). This is far from convincing evidence to demonstrate the reliability and temporal stability of the measurement system. I would required the exact same N = 28 (I'd be happy with N = 27 based on Glüer's [Osteoporos Int. 1995;5(4):262-70. doi: 10.1007/BF01774016.] analysis ) to be satisfied, and retest reliability would have to be measured on different testing sessions, preferably not on the same day. Please move the first part of your study into an appendix to avoid setting a bad precedent (alas, multiple such examples can already be found from the literature but that is no excuse to follow suit).

lines 249 - 261: Please include mean squared error (of the trace that is, not just the events) as a holistic measure of precision (include stats comparisons as well). Exactly like you did with the event-based analyses.

Discussion

lines 333 - 337: I don't think this ought to be your primary aim since your data is much less than convincing for this part. Please consider rephrasing your primary aim (which was to study the sonification). You did, however, show that the IMU-based traces match VICON-based traces in the second part of your study (with N = 28) so you have successfully established concurrent criterion validity.

lines 339 - 342: The options are not only MOCAP or IMUs, why not use e.g., an electronic goniometer? Goniometers could even be considered the golden standard and are affordable as well. Particularly for the knee joint a goniometer would be well-justified as the joint is essentially a hinge joint.

lines 353 -367: I would definitely like to see you contrast your findings against the many previous explorations of MIMU-based systems sagittal plane kinematics concurrent criterion validity. Your results are congruent to the previous literature, a systematic difference is invariably observed, and the difference between systems is heteroschedastic (I noted with pleasure that you had cited Picerno's paper [40] so you definitely are aware of the prior art, I'd just like to see it discussed as well).

Author Response

Thank you for the comments and please see the attachment. 

Reviewer 2 Report

 Item 2.1.2 needs serious revision. The authors are encouraged to rewrite the section starting from the reference systems' definition in a much more informative drawing than Figure 2, which should be completely remade. Then, show the equations used to calculate the knee angle. All terms must be defined contrarily to the p vector in line 134. Figure 2 is very poor, and I don't understand what you mean by "V=Volume".

Considerations on materials and methods:

Page 5:

In the first part of the study, the participants' gait speed was not described, nor was the selection criteria for this speed.

It was also not described why the evaluation contained only two individuals, the selection criterion for these two individuals and their characteristics such as gender, age, height and body mass.

What is the reason for placing participant 1 from 1 to 5 and participant 2 from 6 to 10? Why not put 1 to 5 for both?

The criterion chosen to use a 4km/h gait speed was not described in the second part of the study. Also, data on the participants' body mass, height and gender were not described.

In the third part of the study, the criterion for using a speed of 3.5 km/h was not described.

The authors did not describe how the participants were instructed regarding the sonification feedback; details about the questionnaire were not presented, including whether it was validated and how many questions it had.

Page 6:

The results section presents information that should be in the materials and methods section, paragraph starting on line 264.

Page 8:

It presents information that should be in the materials and methods section, paragraph starting on line 292.

Page 9:

In table 4, which should be table 3, the procedure to obtain the result from the Likert scale and what these values represent was not described.

 Considerations on discussion and conclusion:

Lines 193-199; 222-227 5 minutes or 10 seconds collected and analyzed? 

What are the average values of anthropometric data, such as height and weight?

Was the group composed of individuals of both sexes?

There is no paragraph indentation in lines 136, 143, 264, 271, 277, 292, 297, 307, 318

Line 107, Figure 1a is not cited in the text.

Lines 144-145, "The program provides is (remove IS) a continuous real-time OPTICAL (you mean VIDEO? This is not about optics, to be precise) display of the knee angle trajectories" 

Line 158, "In this case(,) knee angle refers to the (literature declaration) of the sagittal knee angle", you mean refers to the READING of the sagittal knee angle"?

Lines 165-171 – what are A3, A4 and A6????

Line 216, what is S1?

Line 255, did you check for the normality of data before choosing SPM (and not SnPM)?

Lines 255 and 308, where is Figure S3?

Line 259, did you check for non-normality before using the Friedman test?

Lines 264-268, this paragraph should appear before line 249 since you used the variable and defined it afterward.

Table 2, it is not usually necessary showing both SD and variance. Otherwise, are the distributions normal? Otherwise, you should report median and IQR.

Line 308 - For the 'continious'

Lines 318-319, were these data obtained with the knee brace?

Line 322: There is no table 3, although yes, Table 4.

Line 347, the term 'moment' doesn't make sense in this phrase.

Line 399, 'A major limitation is, that'… No comma after 'is'.

Line 409 developement

Line 417, I definitively found no supplementary material in the submission.

Author Response

(The authors gave the same response as above.)

Round 2

Reviewer 1 Report

I am happy with the revised version, congratulations on the interesting work! I did note a few typos, e.g. 'independent meaurement units' in the intro, should read 'inertial', and 'alighted' should read 'aligned' in the methods related to the initialisation procedure. I suspect typing errors will be addressed by the editorial staff in the proofs.

Author Response

Thank you very much for the encouraging words. We are glad that you are satisfied with the revised version. We have only corrected the typos and made a few changes to improve the readability of the text. Based on another reviewer's comment, we have removed the Bosh data sheet from the supplementary material and added it to the references. 

Thank you for taking the time and best regards. 

Reviewer 2 Report

I think it doesn’t seem appropriate including Bosh sensor datasheet as a supplementary material.

Line 266 “inceraasement”

Please include a reference regarding the quaternion matrices calculation.

Author Response

Thank you for your comments. We are pleased that you seem much happier with the revised manuscript. We have removed the Bosh data sheet from the supplementary material. We have added the information as reference [44]. We can only provide the reference for the quaternion calculation of the MIMUs, as the rest is a routine written by the technician and is described in the manuscript according to his information. We hope that you agree with this.

Thank you very much for your effort and best regards.